# Precise distance measurement with stereo camera: experimental results

Haydar Yanık[1] and Bülent Turan[2]

[1] Department of Electronics and Automation, Merzifon Vocational School, Amasya University, Amasya, Turkey
[2] Department of Electricity and Energy, Sivas Vocational School of Technical Sciences, Sivas Cumhuriyet University, Sivas, Turkey



## ABSTRACT

Today, image processing is used in many areas, especially artificial intelligence. This is because images are thought to contain a lot of information. In addition, many distance measurement studies have used image processing techniques. However, no studies have reached these high sensitivity and accuracy rates that can be used in engineering. The motivation of the study is to obtain the results of the experimental application of the image processing method, which can measure distances with high sensitivity and can also be used in engineering fields. In the study, the distances of 19 different target objects were measured using Total Station, Laser Meter, and Developed Prototype (Image Meter). Total Station measurement results were used as a reference and the Laser Meter and Image Meter results were compared. As a result of the comparison, it was determined that the developed Image Meter had a smaller error rate in 11 of the 19 comparisons. Results were obtained with an average error of 1.24% as a result of 19 measurements made with the developed Image Meter. The experimental results were also compared with theoretical calculation. As a result of the comparisons, it was determined that the results with the developed Image Meter were acceptable and could be improved with mechanical arrangements.

## INTRODUCTION

Image processing applications are used in many areas today. The image's pixel values are the basic image data that contain a lot of information (*Kurnaz & Gül, 2018*; *Kılınç & Gözde, 2020*; *Ülkir, Ertuğrul & Akkuş, 2021*; *Selçuk, Çolakoğlu & Alkan, 2018*), including the distance information of the object to the camera. Distance information is basic information required for many systems (*Takatsuka et al., 2003*; *Phelawan, Kittisut & Pornsuwancharoen, 2012*; *Seshadrinathan et al., 2017*; *Adil, Mikou & Mouhsen, 2022*). An inference can be made about the physical properties of the target object according to its distance (*Duran & Turan, 2022*). There are many systems for industrial distance measurement. Physical distance meters, distance meters using the propagation of laser/lidar/radar-based beams and image processing-based distance meters are preferred for daily or industrial work. Distance measurement methods and system preferences vary depending on the object and process. In these preferences, sometimes it is desired that

Corresponding author
Bülent Turan,
bulentturan@cumhuriyet.edu.tr

there is no contact between the target object and the measurement system. In such cases, distance measurement systems based on beam propagation or image processing are preferred for distance (*Yoldaş & Sungur, 2020*; *Yanik & Turan, 2023*; *Cao et al., 2013*). However, to make precise/exact measurements with image processing-based distance measurement systems, the target object image must be obtained in a noiseless and stable manner. In noisy environments (rainy-foggy weather conditions), and in cases where the object image cannot be obtained stably during the measurement period (movement of the target object, sudden changes in the illumination level during the measurement), it is not possible to make precise/exact measurements. In the conducted study, measurement results were obtained by providing suitable conditions. Also, the measurement tolerances of the distance are as important as the accuracy of the distance measurement. This tolerance should be within the acceptable measurement range (*Yanik & Turan, 2021*).

In image processing-based distance measurement systems, mono camera or stereo camera images are used (*Srijha, 2017*). In distance measurement with mono camera images, the distance of the target object is calculated with additional equipment or images from different positions (*Yanik & Turan, 2023*; *Süvari et al., 2021*; *Yamaguti, Oe & Terada, 1997*). In addition, as a different method, the depth maps with mono camera images (*Mayer, 2016*) and the distance information from these maps provide information about the relative position of two points on the image (*Robert & Deriche, 1996*).

The other method, distance measurement with stereo camera images, is based on the registration of image samples taken from two different locations and the calculation of the position of the target object in the image (*Duran & Turan, 2022*; *Tsung-Shiang, 2015*).

Fixed focal length cameras are used in distance measurement with both mono cameras and stereo cameras. For fixed focal length cameras, the field of view of the camera and the number of vertical and horizontal pixels covered by the target object within this field are important (*Azouz, Asli & Khan, 2023*). This data is related to the lens angle of the camera (FoV, field of view) (*Kumar et al., 2023*). At the same time, this data affects the number of pixels per meter (PPM, pixel per meter) (*Yanik & Turan, 2023*; *Payawal & Kim, 2023*). The reason why distance detection systems made with image processing cannot produce high accuracy and stable results is that the PPM value changes depending on the distance (*Phelawan, Kittisut & Pornsuwancharoen, 2012*; *Ashoori & Mahlouji, 2017*; *Hassan et al., 2017*). When stereo images are used, different image thresholding methods can be preferred for similarity rates, and comparisons can be made (*Elen, 2020*).

In this study, the distance between two camera lenses and the angles of these cameras relative to each other were used as the most important features in the distance measurement system with a stereo camera (*Theodosis, Wilson & Cheng, 2014*; *Gan et al., 2018*; *Liang, 2018*). The measurement limits of the system were increased due to the fact that the position angles of the cameras positioned in two different positions and the distance between the cameras lens (A Camera-B Camera) were variable thanks to an electromechanical system (*Katada, Chen & Zhang, 2014*; *Oh et al., 2013*). When the angle and distance information between the camera lenses are obtained accurately and precisely is processed trigonometrically, the distance can be calculated precisely (*Yanik & Turan, 2021*).

In the study, one fixed and the other angular (variable camera angle) and linearly movable (variable distance between lenses) cameras were used. These camera movements were provided synchronously by an electromechanical system. More detailed information about image acquisition, target object selection and system operation is given in the "Material and Method" section. Experimental results obtained by this system and theoretically calculated results are given in the "Experimental Results" section.

## MATERIALS AND METHODS

The system designed for the stereo distance meter is given in Fig. 1. In the system, Camera A is fixed and is the reference camera where the target object is determined and selected. Camera B can move angularly on its own axis and linearly on the central axis. The electromechanical mechanism that provides the angular movement of Camera B can move angularly at $(6.8 * 10 - 4)0$ degrees (526,374 pulses/revolution, Dynamixel PM42-010-S260-R) (*Kim & Oh, 2024*). Its linear movement is 1 mm sensitive.

In the study, the Nikon DTM 362 Total Station distance measuring device was used to measure the actual distances of target objects. Nikon DTM 362; ± (3 + 2 ppm x D) mm distance measurement accuracy has the same distance accuracy for both prisms and reflector sheets and can measure with 1 mm distance resolution in just 1.6 s. A sample image of a total station is given in Fig. 2.

In addition, a G-sniper 1,000 m laser distance meter was used to compare the image meter developed in the study. The G-sniper laser meter image is given in Fig. 3.

### The method used in the experimental study

The triangulation method is used in distance measurement studies conducted with stereo cameras in the literature. In the triangulation method, the measurement error increases non-linearly depending on the distance. The non-linear increase in the measurement error as the distance increases is due to the pixel-based operation of the triangulation method. The motivation of the study is the non-linear increase in the measurement error due to the triangulation method. With this motivation, it is thought that the triangulation method is a pixel-dependent method and that distance measurement should be freed from pixel dependency. The triangulation method in the literature is given in Fig. 4.

In the triangulation method in the literature, the short side information in the right triangle equation is found with the pixel difference data and used. In this study, the method proposed by *Yanik & Turan (2023)* as the improved triangulation method was used. In the improved triangulation method, the real measure of the short side information is found. This allows the distance to be found precisely. The improved triangulation method consists of two stages.

In the first stage, the moving camera is moved angularly and thus the target object location in the camera images is matched. This ensures that the pixel difference used in the triangulation method is 0. However, having a pixel difference of 0 does not guarantee that the image acquisition axes of the cameras intersect on the target object. Because pixels cover an area. In other words, they have a width. For this reason, the distance cannot be

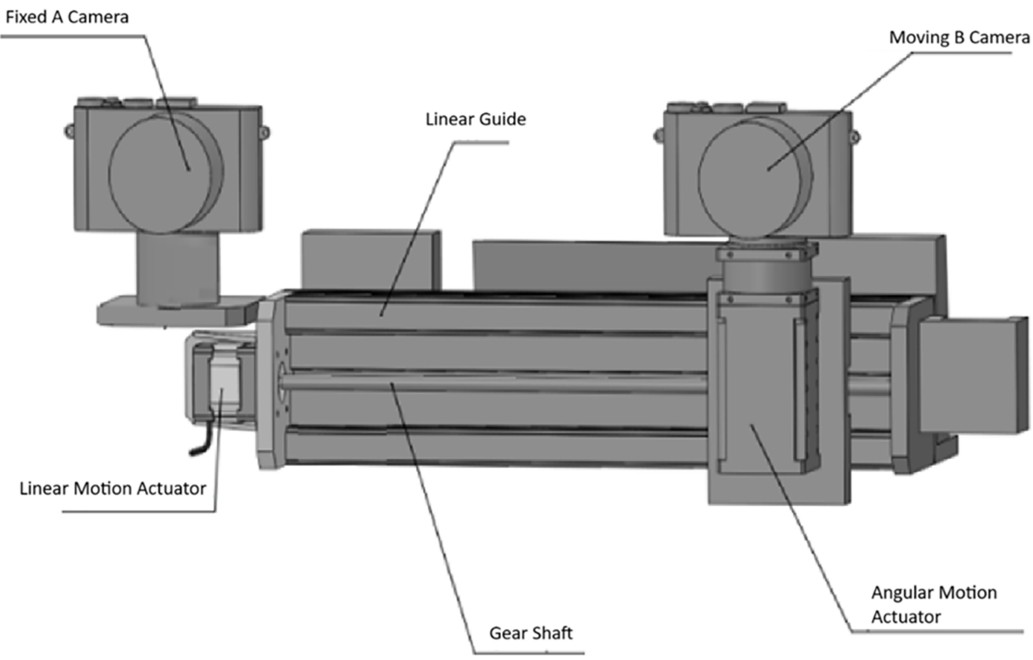

**Figure 1 Designed stereo rangefinder (*Yanik & Turan, 2023*).**

measured precisely/exactly in a measurement calculation made using only the angle. In the second stage of the improved triangulation method, the moving camera, which has completed the angular movement, is moved linearly on the horizontal axis. With the linear movement, the distance between the lenses that ensure that the image acquisition axes of the cameras coincide on the target object is determined. This distance is also the short side length of the right triangle to be used in the triangulation method. And its precise/exact determination also ensures that the long side value (object distance) is found precisely/exactly. The details of the method are given under the title of performing the measurement process.

## Performing the measurement process

A target is determined by the user in the reference camera screen image. Any pixel belonging to the target object is marked by the user with the help of a marker. The marked pixel and this pixel are considered the center, and a $71 \times 71$ pixel template is created around them. This template is a 3D matrix of $71 \times 71$ pixels. This matrix is reduced to binary image level and single dimension. The entire screen image belonging to the moving camera is 3D, and the entirety is $4,608 \times 3,456$ pixels. This image is reduced to binary image level and single dimension. The template is started to be searched in Camera B image screen. The similarities between the found image set and the target image set are compared, and a coefficient is determined. Thanks to this coefficient, the search image (camera B image screen) is reduced to $4,608 \times 71$ pixels. The target image search is now performed using the newly created search image. The target image is moved with angular

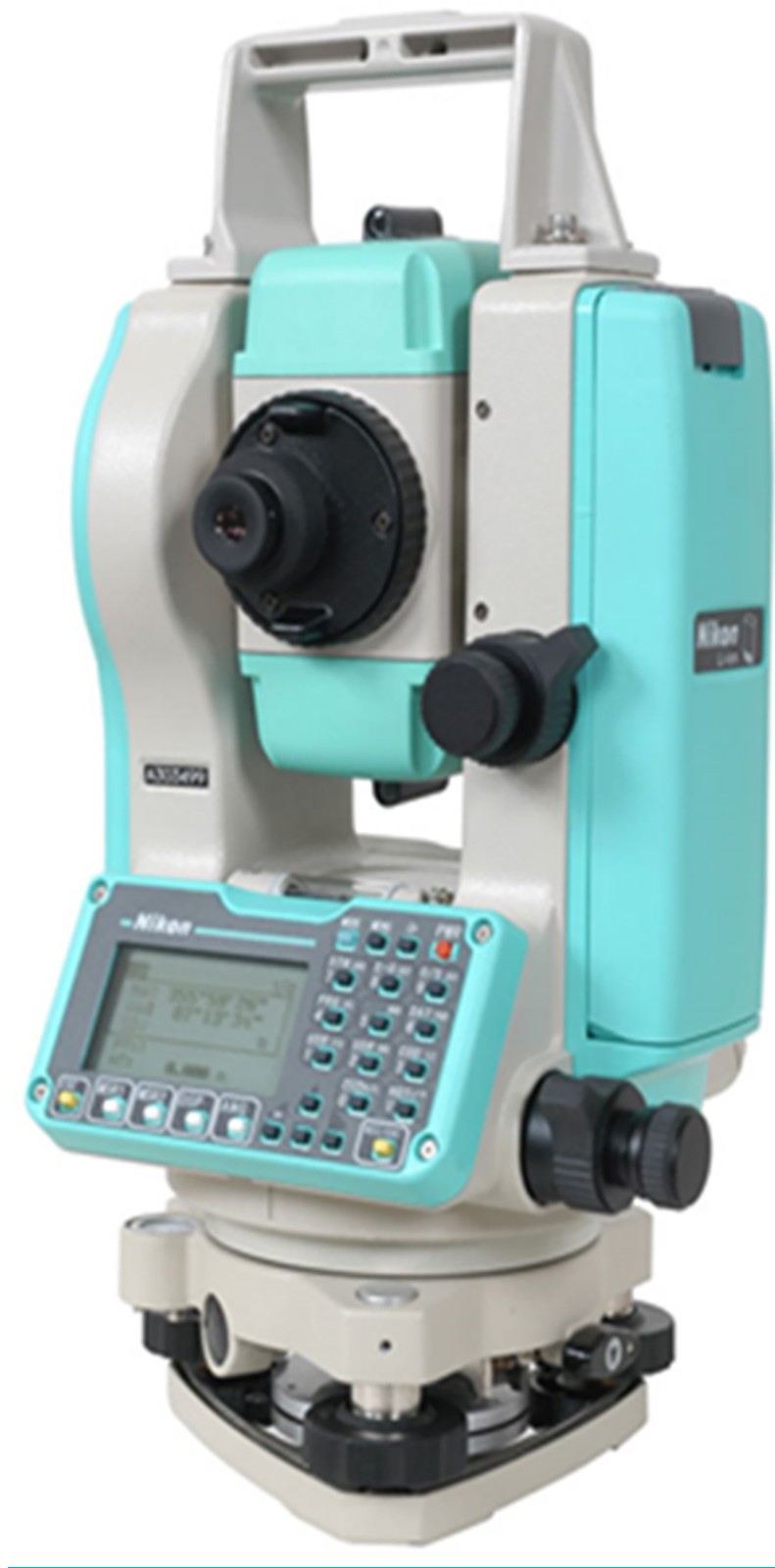

**Figure 2  Nikon DTM 362 total station.**   

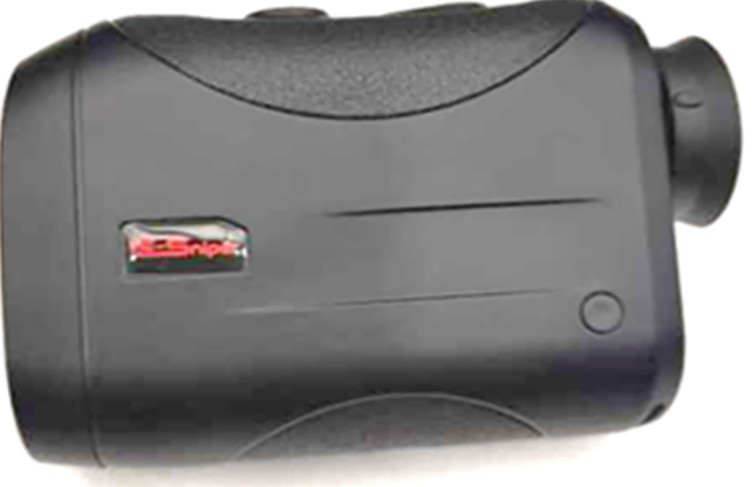

**Figure 3** G-sniper laser meter.

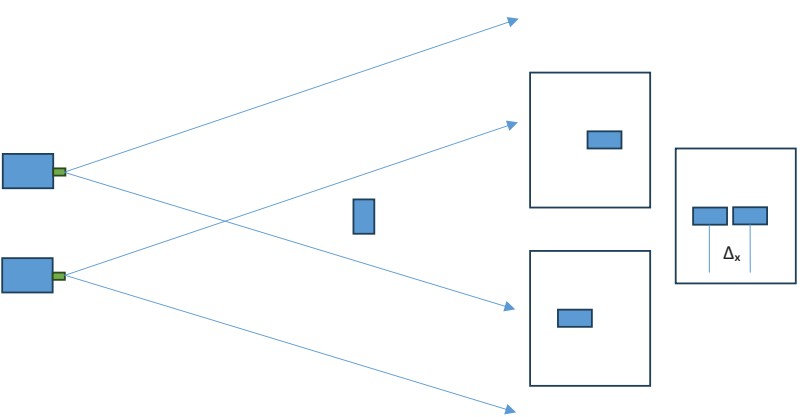

**Figure 4** Pixel shift in triangulation method.

movement until the difference between the centre pixels in the found image falls below 0 pixels. The last angle that makes the difference greater than zero will be the B camera angle. Then, the linear movement of the system will start by keeping this angle fixed. Here, the moving camera is first moved away from the reference camera (in the l+ direction), and the camera position that makes the pixel difference between the two images −1 and, therefore, the distance between the moving camera and the fixed camera lenses is found. Then, the moving camera moves to the other side (in the l-direction), and the camera coordinate makes the pixel difference between the two images 1 pixel. Therefore, the distance between the moving camera and the fixed camera lenses is obtained by the electromechanical system. The electromechanical system trigonometrically calculates the distance of the target object to the system by using the average of the two-distance information and the angle information. The system algorithm (*Yanik & Turan, 2023*) is given in Algorithm 1, and calculation details are given below.

In Eq. (1), the field of view (FoV) of a camera with a viewing angle of $\alpha$ at a distance of Z (m) is calculated.

$$FoV = tan\left(\frac{a}{2}\right) * Z * 2 \tag{1}$$

In Eq. (2), the number of pixels per meter in a camera with a field of view FoV is calculated. The horizontal resolution of the camera is taken as wp.

$$PPM = \frac{wp}{FoV} \tag{2}$$

In Eq. (3), the position difference of the target object in images belonging to cameras at different positions on the same axis is calculated in pixels.

$$\Delta x = PPM * l \tag{3}$$

In Eq. (4), the equation used in calculating the distance in a stereo camera rangefinder, where the distance between the lenses is determined as $l_g$ and the moving camera angle is $\beta$, is given.

$$Z = \frac{l_g}{tan\beta} \tag{4}$$

If the moving camera angle and the distance between the lenses in Eq. (4) are taken from the electromechanical system with precision, the distance information can also be calculated with that precision.

Eqs. (1), (2), (3), (4) are the equations used in distance information and processing this information.

In stereo camera distance detection studies, pixels must, of course, be present, and when two different cameras with different positions are used, the same object will be displayed in different positions in both cameras. However, by changing the camera's image acquisition axis angles, the object images can be moved to the same position.

The study focused on this. In our study, fixed (A) and mobile (B) cameras were used at different points. The image taken from camera B was superimposed with the image taken from the reference camera (Camera A). Thus, the difference between the target object positions in the camera screen images can be changed with the angular movement of camera B. If the image acquisition axis angle of camera B and the distance between the lenses can be precisely determined at the moment when this difference is 0, the distance of the object to the cameras can be precisely determined.

The overlapping process of the object images formed on cameras A and B is provided by the angular and linear movements of the mobile camera B. First, the distance between the lenses is kept constant (0.25 m). Camera B is moved angularly, and $\Delta x = 0$ is captured. Thus, the angle $\beta$ is found. Then, the system starts the movement between the lenses by keeping the $\beta$ angle constant. The distance between the lenses, which is 0.25 m, is increased by 0.001 m and the $l_+$ that provides the condition $\Delta x > 0$ on the camera's image is received by the system. Then, the lenses come back to their initial position of 0.25 m and the second movement between the lenses begins, which is the reduction process of 0.001 m. The value that makes $\Delta x < 0$ becomes the $l_-$ value of the system. This situation finds how much

**Algorithm 1 Distance determination process algorithms with image processing.**

Input Values:          A Camera Image (Reference Image), B Camera Image

Output Values:        Hedef Nesne Mesafesi ($Z$)

Initial Values:        $\beta = 0^0$,      $l = 0.250$ m

If the target object is marked in the reference image

    Distinguish target object from background image

    Find the target object in camera B image and distinguish it from the background image

    $\Delta x$ calculate

    ($\Delta x > 0$) As long as the condition is met

        *Increase the $\beta$ angle by 1 step (0.001°)*

      $\Delta x$ calculate

    End

    *Record the angle $\beta$*

    ($\Delta x < 1$) As long as the condition is met

        *Increase the $l$ value by 1 step (0.001 m)*

      $\Delta x$ calculate

    End

    *Record the $l_+$ value*

    Return the initial $l$ value (0.250 m)

    ($\Delta x > -1$) As long as the condition is met

        *Decrease the $l$ value by 1 step (0.001 m)*

      $\Delta x$ calculate

    End

    *Record the $l_-$ value*

    Calculate and save $l_g$ value

    Calculate Distance ($Z$)

If no conditions are met

    Back to top

End

distance the width corresponds to linearly between −1 and +1 pixels on the image screens of the cameras. The average value will be the midpoint of these two values. This value is the real distance between the lenses that we will use in the trigonometric calculation. Then, the angle and distance information between the lenses obtained electromechanically are substituted in "Eq. (4)", and the system calculates the actual distance. The image meter theoretical diagram is given in Fig. 5.

In Fig. 5, the distances of 1 pixel belonging to the target object on the image screens and the distance between these distances are given symbolically. The distance between the two cameras is initially 0.25 m, and the angles of the two cameras are positioned parallel to each

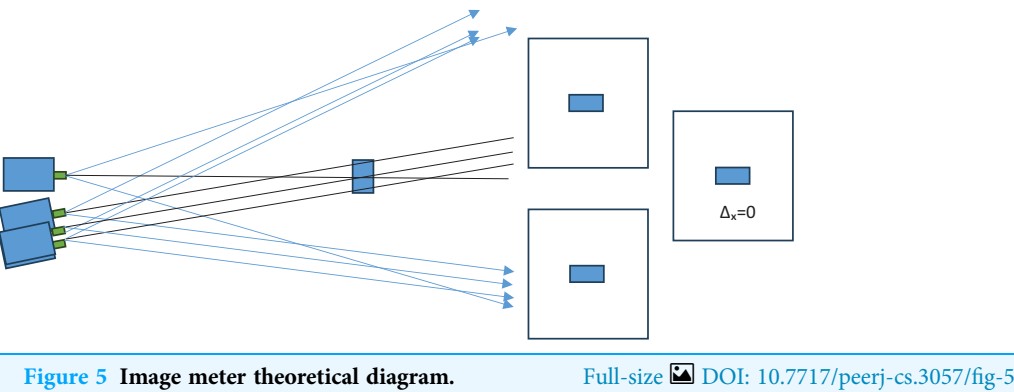

**Figure 5   Image meter theoretical diagram.** 

other. The user marks the target with a marker from the screen accepted as the reference image. The pixel selected by the user represents a unit belonging to the target object. A $71 \times 71$ matrix is created with this selected pixel and neighbouring pixels. Then, this $71 \times 71$ matrix is searched in the target image. This first part, which is outside the vertical axes, is removed from the image. The search is now made in $4{,}608 \times 71$ pixels. In the following steps, the search continues by narrowing down the searched image set. In the last stage, the template searched for is found in the image. Thus, the location of the target object is determined on both image screens. In the initial state, the angular movement of the moving (B) camera starts until the difference between the two positions becomes 0 in the image screens belonging to the two cameras that are 0.25 m away from each other. The angle value ($\beta$) that makes $\Delta x = 0$ will be the angle value of the system. Then the system is moved linearly so that the first value $l_+$ that makes $\Delta x > 0$ and the first value $l_-$ that makes $\Delta x < 0$ are calculated.

$$l_g = \frac{(l_+) + (l_-)}{2}. \tag{5}$$

The $l_g$ value calculated according to Eq. (5) becomes the real l value of the system. Then, the $l_g$ and $\beta$ values obtained by the system are substituted into Eq. (4) and the real distance is calculated.

The target object sketch and capture of the target object image with the image meter are given in Fig. 6.

In Fig. 6, the distance diagram between the target object and the platform and the images of the target object are given.

## EXPERIMENTAL RESULTS

In the experimental study, distances were measured with Total Station and these distances were accepted as the real distance (reference). The reason why Total Station measurements were accepted as the reference is that these devices are widely used in the field of engineering and the measurement results are accepted as the real measurement value. A laser meter was used to compare the measurement results in the study.

The prototype developed for the open-field experimental study was set up in front of the window in the researcher's office. Nineteen targets with different distances visible from the

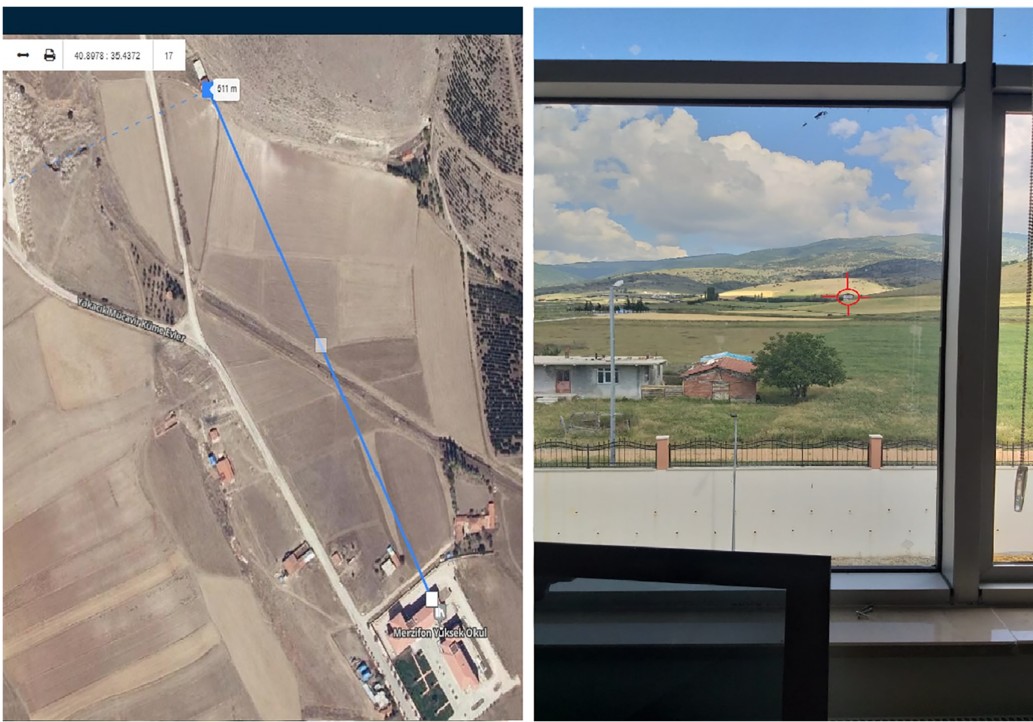

**Figure 6 Target object sketch and capture of target object image with image meter.**

window were determined. The determined target distances were measured and recorded with a Total Station device and laser meter before the marked measurement point. Then, each object's distance was measured and recorded with the developed prototype. The reason for choosing the researcher's office for the establishment of the prototype was to create a standard and accessible experimental area and to ensure the sustainability of the power supply of the prototype. The determined target objects were fixed (non-moving) objects such as buildings, huts, and trees. Thus, the same object distance could be measured reliably at different time periods during the experiment.

## Determining the angle of the moving camera

The moving camera, moves angularly until the locations of the target object (the specified template representing the target object) in the two images (reference and moving camera images) coincide. When the target object locations coincide, the angular movement is stopped, and the moving camera angle is recorded. During the determination of the moving camera angle, the distance between the lenses is kept constant at 0.25 m. The angle information in the measurements is given in Table 1. Since the values given in Table 1 are calculated with the fixed distance information between the lenses, the error rates are high.

The distance information calculated with the distance between fixed lenses and the specified angle information is not precise. Because pixels in the image have an area (horizontal and vertical dimensions). When the moving camera takes an image with the specified angle, even if the image acquisition axis coincides with any part of the horizontal

**Table 1 Measurement values and calculated error rates obtained using only the β angle.**

| Actual distance | $l_b$ | β-measured | Z | Error |
|---|---|---|---|---|
| 6.723 | 0.25 | 1.7425487 | 8.2175771 | 22.23% |
| 10.412 | 0.25 | 0.9894593 | 14.475099 | 39.02% |
| 16.288 | 0.25 | 0.8754822 | 16.359933 | 0.44% |
| 21.054 | 0.25 | 0.6837542 | 20.947974 | 0.50% |
| 35.268 | 0.25 | 0.3756421 | 38.131348 | 8.12% |
| 48.271 | 0.25 | 0.2472702 | 57.927951 | 20.01% |
| 48.796 | 0.25 | 0.2375488 | 60.298612 | 23.57% |
| 55.669 | 0.25 | 0.2437845 | 58.756231 | 5.55% |
| 66.322 | 0.25 | 0.1698535 | 84.330926 | 27.15% |
| 160.578 | 0.25 | 0.0866421 | 165.32302 | 2.95% |
| 225.617 | 0.25 | 0.0698754 | 204.99257 | 9.14% |
| 276.274 | 0.25 | 0.0602155 | 237.87795 | 13.90% |
| 355.363 | 0.25 | 0.0363524 | 394.03019 | 10.88% |
| 368.634 | 0.25 | 0.0295649 | 484.49153 | 31.43% |
| 489.651 | 0.25 | 0.0276355 | 518.31679 | 5.85% |
| 511.367 | 0.25 | 0.0224588 | 637.7876 | 24.72% |
| 563.972 | 0.25 | 0.0227246 | 630.32767 | 11.77% |
| 645.866 | 0.25 | 0.0202875 | 706.04778 | 9.32% |
| 801.955 | 0.25 | 0.0165754 | 864.16886 | 7.76% |

edge of the pixel, the object images coincide with the information according to the target object location calculation. However, for precise distance measurement, the image acquisition axes of the moving camera and the fixed camera must coincide at the centre point of the specified pixel on the target object (the specified pixel belonging to the target object). In order to ensure this, the moving camera is precisely moved linearly on the horizontal axis, keeping the specified angle constant. The optimum distance between lenses is determined, and the calculation is made based on these two features.

When the data obtained in the table was evaluated, it was seen that the error rates in the measurements were variable. However, a linear increase was not observed depending on the distance. These measurements were the first stage of the experiment only as a result of the angular movement of the system. After each measurement, these angle values were kept constant, and the distances between the lenses were found again.

For example, In Table 1, the measured β angle value for the target distance at 6.723 m, which was measured by keeping the distance between the lens's constant (lb = 0.25 m), was 1.74254870 degrees. When the system processed this data, the distance was approximately 8.21m, and the margin of error was 22.23%. When the angle information measured in Table 1 was kept constant for the same distance, and the recalculated distance between the lenses was taken into account, the result would be as in Table 2. In other words, when the β value for the target at 6.723 m was kept constant at 1.74254870, the distance between the lenses was recalculated by moving the moving camera linearly, and the average distance

**Table 2  Measurement results and error rates obtained by using β and lg values together.**

| Actual distance | Parameters obtained by the electromechanical system | | | | Measurement results (m) | Error % | L = 0.25 m fixed Error % |
|---|---|---|---|---|---|---|---|
| | $l_+$ (mm) | $l_-$ (mm) | $l_g$ (mm) | $\beta$ | | | |
| 6.723 | 246.85 | 166.12 | 206.485 | 1.742548675 | 6.787 | 0.96 | 22.2 |
| 10.412 | 241.54 | 115.24 | 178.390 | 0.989459280 | 10.329 | 0.80 | 39.0 |
| 16.288 | 289.75 | 209.54 | 249.645 | 0.875482155 | 16.337 | 0.30 | 0.44 |
| 21.054 | 257.46 | 238.54 | 248.000 | 0.683754210 | 20.780 | 1.30 | 0.50 |
| 35.268 | 267.84 | 203.56 | 235.700 | 0.375642103 | 35.950 | 1.93 | 8.12 |
| 48.271 | 242.85 | 176.75 | 209.800 | 0.247270173 | 48.613 | 0.71 | 20.0 |
| 48.796 | 240.65 | 158.05 | 199.350 | 0.237548754 | 48.082 | 1.46 | 23.6 |
| 55.669 | 263.57 | 201.12 | 232.345 | 0.243784513 | 54.607 | 1.91 | 5.55 |
| 66.322 | 244.51 | 145.43 | 194.970 | 0.169853452 | 65.768 | 0.84 | 27.1 |
| 160.578 | 297.62 | 196.86 | 247.240 | 0.086642104 | 163.500 | 1.82 | 2.95 |
| 225.617 | 304.53 | 233.54 | 269.035 | 0.069875422 | 220.600 | 2.22 | 9.14 |
| 276.274 | 255.55 | 318.54 | 287.045 | 0.060215487 | 273.130 | 1.14 | 13.9 |
| 355.363 | 318.67 | 135.67 | 227.170 | 0.036352423 | 358.005 | 0.76 | 10.9 |
| 368.634 | 242.65 | 135.80 | 189.225 | 0.029564875 | 366.710 | 0.52 | 31.4 |
| 489.651 | 318.47 | 146.37 | 232.420 | 0.027635486 | 481.870 | 1.59 | 5.85 |
| 511.367 | 248.56 | 154.25 | 201.405 | 0.022458762 | 513.810 | 0.48 | 24.7 |
| 563.972 | 317.42 | 138.45 | 227.935 | 0.022724579 | 574.690 | 1.90 | 11.8 |
| 645.866 | 308.97 | 154.24 | 231.605 | 0.020287541 | 654.090 | 1.27 | 9.32 |
| 801.955 | 306.82 | 165.24 | 236.030 | 0.016575422 | 815.870 | 1.74 | 7.76 |

between the lenses was found, $l_g$ = 206.485 mm. When the β calculated in Table 1 and $l_g$ calculated in Table 2 were taken into account again, the real measurement result would be 6.787 m. In this case, our margin of error will decrease from 22.23% to 0.96%. The low margin of error for some measurements in Table 1 was coincidental for the calculated distances. In these measurements, the moving camera image acquisition axis coincided with a region close to the centre point of the target pixel.

As another example, the β value in Table 1 for the distance of 801.955 m was found to be 0.01657540 degrees, and when lb = 0.25 m was taken, the calculated distance was calculated as 864.16886 m. Our margin of error here was 7.76%. If we look at the measurement values in Table 2 for this distance, the $l_g$ value measured in the second case was 236.030 mm, and the distance measured by the system was 815.87 m. Our margin of error in this measurement is also 1.74%. The distances between the lenses, which were recalculated by keeping the β angle in Table 1 constant, the measurement results of the electromechanical system and the margins of error are given in Table 2.

When the measurement errors made at the fixed lens distances measured by the electromechanical system are compared, a significant decrease in the margin of error is observed. This is due to the precise acquisition of the real angle and the distance between the lenses required to find the distance information.

**Table 3 Comparison of image meter and laser meter measurement results.**

| TS (m) | LM (m) | Parameters measured with GM | | | | GM (m) | Error TS/IM % | TS/LM % |
|---|---|---|---|---|---|---|---|---|
| | | $l_+$ (mm) | $l_-$ (mm) | $l_g$. (mm) | β (Angle) | | | |
| 6.723 | 6.8 | 246.85 | 166.12 | 206.485 | 1.742548675 | 6.787 | −0.96 | −1.15 |
| 10.412 | 10.3 | 241.54 | 115.24 | 178.390 | 0.989459280 | 10.329 | 0.80 | 1.08 |
| 16.288 | 16.5 | 289.75 | 209.54 | 249.645 | 0.875482155 | 16.337 | −0.30 | −1.30 |
| 21.054 | 20.8 | 257.46 | 238.54 | 248.000 | 0.683754210 | 20.780 | 1.30 | 1.21 |
| 35.268 | 35.6 | 267.84 | 203.56 | 235.700 | 0.375642103 | 35.950 | −1.93 | −0.94 |
| 48.271 | 48.8 | 242.85 | 176.75 | 209.800 | 0.247270173 | 48.613 | −0.71 | −1.10 |
| 48.796 | 47.6 | 240.65 | 158.05 | 199.350 | 0.237548754 | 48.082 | 1.46 | 2.45 |
| 55.669 | 53.6 | 263.57 | 201.12 | 232.345 | 0.243784513 | 54.607 | 1.91 | 3.72 |
| 66.322 | 65.4 | 244.51 | 145.43 | 194.970 | 0.169853452 | 65.768 | 0.84 | 1.39 |
| 160.578 | 165.8 | 297.62 | 196.86 | 247.240 | 0.086642104 | 163.498 | −1.82 | −3.25 |
| 225.617 | 218.8 | 304.53 | 233.54 | 269.035 | 0.069875422 | 220.601 | 2.22 | 3.02 |
| 276.274 | 278.3 | 255.55 | 318.54 | 287.045 | 0.060215487 | 273.127 | 1.14 | −0.73 |
| 355.363 | 358.8 | 318.67 | 135.67 | 227.170 | 0.036352423 | 358.047 | −0.76 | −0.97 |
| 368.634 | 370.2 | 242.65 | 135.80 | 189.225 | 0.029564875 | 366.712 | 0.52 | −0.42 |
| 489.651 | 486.7 | 318.47 | 146.37 | 232.420 | 0.027635486 | 481.869 | 1.59 | 0.60 |
| 511.367 | 512.6 | 248.56 | 154.25 | 201.405 | 0.022458762 | 513.815 | −0.48 | −0.24 |
| 563.972 | 565.6 | 317.42 | 138.45 | 227.935 | 0.022724579 | 574.695 | −1.90 | −0.29 |
| 645.866 | 659.8 | 308.97 | 154.24 | 231.605 | 0.020287541 | 654.095 | −1.27 | −2.16 |
| 801.955 | 810.5 | 306.82 | 165.24 | 236.030 | 0.016575422 | 815.878 | −1.74 | −1.07 |

(TS/IM)
- Absolute standard deviation of percentage error, 0.578
- Standard deviation of percentage error, 1.403
- Absolute mean error, 1.245
- Mean error, −0.005
- Confidence intervals 95%* −0.73 < μ < 0.63

(TS/LM)
- Absolute standard deviation of percentage error, 1,015
- Standard deviation of percentage error, 1.782
- Absolute mean error, 1,426
- Mean error, −0.008
- Confidence intervals 95%* −0.94 < μ < 0.78

**Note:**
TS, Total Station; IM, Image Meter; LM, Laser Meter, Error rates were calculated for IM and LM using the TS measurement device as reference. *For significance determination, mean error was calculated with absolute values. Standard deviation and confidence interval were calculated with real values.

When the results from measurements made at different distances are examined, it is seen that the margin of error is different from the systems found in the literature and does not increase with distance.

The results of the image meter measurement device developed in our study were also compared with the results of the laser meter measurement device. In the comparison phase, the results of the Total Station measurement device were used as a reference. The comparison results are given in Table 3.

Table 3 shows the GM, LM and TS measurement values of the same target. According to GM, the largest error was detected at 225,617 m with a 2.22% error rate. The best results were obtained at 16,288 m with 0.30% and 511,367 m with 0.48%.

In addition, the errors mean were determined. The standard deviations and confidence intervals of the determined error means were also determined. Since the errors included negative and positive results, the meaning of the absolute error values was taken into account in terms of significance. However, real values were used in determining the standard deviation and confidence interval. For GM, the error mean was found to be 1.245%, the standard deviation as 1.403 and the confidence interval as −0.73 < μ< 0.63. For

LM, the error mean was found to be 1.426%, the standard deviation as 1.782 and the confidence interval as $-0.94 < \mu < 0.78$. When both the errors mean, the standard deviation and the confidence interval were taken into account, GM had better results than LM.

Target object sizes were selected to be visible to the camera. The measurements were made by determining the objects in the natural environment. For this reason, the measurement distances do not have a standard range. And they do not consist of exact values.

## DISCUSSION

No studies in the literature aim to determine precise distances with image processing. In the studies in the literature, it is seen that the measurements are made at short distances and that the error rates increase parabolically as the distance increases despite being made at short distances. The literature studies show that the longest distance is limited to 100 m, and the error rate is very high. These studies aim to determine the precise distance when measurements are made in closed environments and at short distances. However, it has been seen that the studies do not aim to develop a measurement device that is planned to be used in engineering. The expected results to be obtained when the triangulation method is used at long distances are calculated and given in Table 4.

Table 4 shows the pixel differences to be obtained when the triangulation method in the literature is used in short and long distances and the error rates to occur in consecutive measurements. In addition, as can be clearly seen in Table 4, even if the camera resolutions are high, the pixel difference becomes 0 after a certain distance. This means that distance measurements cannot be made after a certain distance even if the resolution is high in pixel-based measurement methods.

This study aims to develop a prototype of a measurement device that can be used in the engineering field. In this respect, it differs from other studies in literature.

There is a study on precise distance determination with image processing in the literature. This study is a theoretical study on precise distance determination conducted by the same researchers (*Yanik & Turan, 2023*). As a result of the theoretical calculations made in the study, distance measurement can be made with an error rate of 0.17% for a distance of 900 m. The same study calculated the error rate as 0.07% for 1,000 m. It is understood from this that although the error rate is proportional to the distance, the angular resolution also affects the error rate, and sometimes, the error rate decreases due to the angular resolution as the distance increases.

However, these success rates could not be achieved in the experimental study. The reason for this is thought to be software errors (errors occurring in the image processing phase) and hardware/mechanical errors (errors occurring in the movement of the electromechanical system).

Table 5 shows the comparison table of the results by theoretical calculations with the experimental results.

When Table 5 is examined, it is seen that although the error rates in the theoretical calculation are low, the error rates in the experimental study are high. As a result of

**Table 4 In the triangulation method, the effect of distance on measurement error (*Yanik & Turan, 2023*).**

| Z (m) | FoV (m) | Pixel width (m) | PPM (pixel/m) | Δx = (PPM * l) Fixed l = 0.2 m | Pixel difference between consecutive distances | Max error between consecutive distances (m) |
|---|---|---|---|---|---|---|
| 10 | 11.547 | 0.00250 | 400.000 | 400 * 0.2 = 80 p | | |
| 11 | 12.701 | 0.00276 | 362.310 | 362.31 * 0.2 = 72.462 = 72 p | 8 p/1 m | (1/8) * (1/2) = 0.0625 m |
| 12 | 13.856 | 0.00300 | 333.330 | 66.666 = 67 p | 5 p/1 m | (1/5) * (1/2) = 0.1 m |
| 20 | 23.094 | 0.00501 | 199.600 | 39.92 = 40 p | 13 p/8 m | (8/13) * (1/2) = 0.3075 m |
| 30 | 34.641 | 0.00751 | 133.155 | 26.631 = 27 p | 13 p/10 m | (10/13) * (1/2) = 0.3845 m |
| 600 | 692.820 | 0.15035 | 006.651 | 1.330 = 1 p | 26 p/570 m | (570/26) * (1/2) = 10.9615 m |
| 900 | 1,039.230 | 0.22552 | 004.434 | 0.8868 = 1 p | 0 p/300 m | (300/0) * (1/2) = ∞m |
| 1,000 | 1,154.700 | 0.25058 | 003.990 | 0.798 = 1 p | 0 p/200 m | (200/0) * (1/2) = ∞m |

**Note:**
  * Calculations were made based on images taken with a camera with a horizontal pixel resolution of 4,608.

**Table 5 Comparison of theoretical calculation results with experimental results.**

| TS measurement | Obtained by theoretical calculation | | Measured by image meter | | Error difference |
|---|---|---|---|---|---|
| | Z (m) | Error % | Z (mm) | Error % | |
| 6.723 | 6.719 | 0.06% | 6.787 | 0.95% | 0.89% |
| 10.412 | 10.416 | 0.04% | 10.329 | 0.80% | 0.76% |
| 16.288 | 16.317 | 0.18% | 16.337 | 0.30% | 0.12% |
| 21.054 | 21.047 | 0.03% | 20.78 | 1.30% | 1.27% |
| 35.268 | 35.203 | 0.18% | 35.95 | 1.93% | 1.75% |
| 48.271 | 48.242 | 0.06% | 48.613 | 0.71% | 0.65% |
| 48.796 | 48.614 | 0.37% | 48.082 | 1.46% | 1.00% |
| 55.669 | 55.788 | 0.21% | 54.607 | 1.91% | 1.69% |
| 66.322 | 66.411 | 0.13% | 65.768 | 0.84% | 0.70% |
| 160.578 | 160.602 | 0.01% | 163.498 | 1.82% | 1.80% |
| 225.617 | 223.974 | 0.73% | 220.601 | 2.22% | 1.49% |
| 276.274 | 276.061 | 0.08% | 273.127 | 1.14% | 1.06% |
| 355.363 | 355.928 | 0.16% | 358.047 | 0.76% | 0.60% |
| 368.634 | 368.95 | 0.09% | 366.712 | 0.52% | 0.44% |
| 489.651 | 489.618 | 0.01% | 481.869 | 1.59% | 1.58% |
| 511.367 | 511.755 | 0.08% | 513.815 | 0.48% | 0.40% |
| 563.972 | 563.843 | 0.02% | 574.695 | 1.90% | 1.88% |
| 645.866 | 645.88 | 0.00% | 654.095 | 1.27% | 1.27% |
| 801.955 | 802.141 | 0.02% | 815.878 | 1.74% | 1.71% |

19 measurements through experimental study, an error average of 1.24% was obtained. Of course, it is normal to have software and mechanical system errors in the experimental study. However, the results obtained with the experience of the researchers have determined that mechanical errors are more than software errors in this study. The reason for this is thought to be because the developed prototype was produced on an amateur production bench. It is thought that if the developed device design is produced in a

professional production facility, it is possible to reduce the mechanical system errors that occur greatly.

When the errors that occur/may occur in the image meter device are considered; These errors can be standard and random errors that may occur on the mechanical system, as well as segmentation errors that may occur during the image processing stage. The determined error types are given below and added to the distance calculation equation in Eq. (6) (*Yanik & Turan, 2023*).

$l_{seg}$; *Segmentation error*,
$l_s$; *Standard linear mechanical error*,
$l_r$; *Random linear mechanical error*,
$\beta_s$; *Standard angular mechanical error*,
$\beta_r$; *Random angular mechanical error*.

$$Z = \frac{l_g + l_s + l_r + l_{seg}}{tan(\beta + \beta_s + \beta_r)}. \tag{6}$$

## CONCLUSIONS

When image processing-based stereo camera distance measurement systems are examined, it is seen that the error margins of the systems increase in direct proportion to the distance. In the study, a measurement device prototype independent of pixel errors was developed, and the measurement results of the prototype were given. It was seen that the error average was 1.245% for 19 different measured distances, and the largest error was 225.617 m. When the results of the experimental study are examined, it is seen that the measurement success is very high compared to the studies in the literature. This is sufficient to state that the method developed in the study on precise distance determination with image processing is an effective method. However, the success rate obtained with theoretical calculations could not be reached. This is due to the mechanical hardware used in the developed prototype. In future studies, the success rate will approach the success rate in theoretical calculations by developing mechanical hardware with higher budget projects. Thus, device tests can be made more stable with prototypes with more stable mechanical hardware and measurement errors can be examined in detail.

In addition, studies can be carried out to eliminate standard mechanical errors that may occur during the production phase of the measuring device ($l_s$, $\beta_s$) using calibration methods.

### Funding

This work was supported by Tokat Gaziosmanpaşa University Scientific Research Projects Coordination Unit, project number: 2020/106. The funders had no role in study design, data collection and analysis, decision to publish, or preparation of the manuscript.

## Grant Disclosures

The following grant information was disclosed by the authors:
Tokat Gaziosmanpaşa University Scientific Research Projects Coordination
Unit: 2020/106.

## Competing Interests

The authors declare that they have no competing interests.

## Author Contributions

- Haydar Yanık conceived and designed the experiments, performed the experiments, analyzed the data, performed the computation work, prepared figures and/or tables, and approved the final draft.
- Bülent Turan conceived and designed the experiments, analyzed the data, performed the computation work, prepared figures and/or tables, authored or reviewed drafts of the article, and approved the final draft.

## Data Availability

The code is available in the Supplemental File.

## Supplemental Information

Supplemental information for this article can be found online at http://dx.doi.org/10.7717/peerj-cs.3057#supplemental-information.

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
