# Peer review of "Precise distance measurement with stereo camera: experimental results"

_PeerJ Computer Science, doi:10.7717/peerj-cs.3057_

## Round 0.1 · original submission · Major Revisions

·

Basic reporting

• The manuscript is written in clear and professional English, although minor grammatical corrections are needed in some sections (e.g., singular/plural agreements, prepositions).
• The abstract is concise and informative, effectively summarizing the study's motivation, methods, and findings.
• The background section is thorough, with relevant references cited to justify the novelty of the approach.
• Figures and tables are well-labelled and relevant. Raw data appears to be provided adequately.
• The structure conforms to PeerJ standards.

Experimental design

• The research question is well-defined and meaningful. The study addresses a known gap in engineering-grade distance measurement using image processing.
• The design is original, particularly the electromechanical system that varies the camera angle and baseline to enhance measurement accuracy.
• The experimental protocol is described in sufficient detail to allow for replication.
• The use of Total Station as a reference adds strong credibility to the measurement validation.

Validity of the findings

• The results are clearly presented, and the error metrics are transparent.
• Experimental results show that the developed “Image Meter” has lower average error than a standard laser meter in several cases.
• The method is tested across a wide range of distances, including long-range (up to 800+ meters), which is rarely seen in similar studies.
• Comparison with both theoretical models and commercial tools strengthens the manuscript's reliability.
* The authors acknowledge that mechanical limitations affected experimental precision. This transparency is appreciated, and it is essential to indicate the possible way to improve the precision.
* In the image processing: Using cv2.matchTemplate, which is sensitive to scale, rotation, and noise, try more robust techniques like YOLO or another deep learning model for object detection.

Additional comments

This well-executed experimental study introduces a stereo vision-based distance measurement prototype with impressive performance at long distances. The paper fills a practical gap in the field and demonstrates the real-world viability of a low-cost system. With improved mechanical components and more robust automation (e.g., in template matching), the prototype could become a professional tool in engineering or surveying.

Reviewer 2 ·

Basic reporting

The submitted paper addresses distance measurement through stereo vision. The paper proposes and evaluates experimentally a setup that uses mechanically (automatically) adjustable stereo basis and angle of one of the cameras of the camera pair.

The paper is written in a good English, it contains relevant literature references and provides background information with an overview of existing stereo distance measurement methods. The paper has a solid a feasible structure and includes appropriate figures and tables. The paper is mostly self contained (with an exception mentioned below) and contains experimental results. No “teorems and proofs” are formally included but in this case, this is understandable in that sense that the paper involves an experimental evaluation of the proposed method.

Experimental design

The proposed paper contains original research results within the scope of the journal. The experimental results are generated by a setup proposed and described by the authors and they are well described.

The research questions are formulated indirectly but clearly enough and it fills the "gap" in that respect that the making the stereo based distance measurement more precise definitely is useful for many applications.

I do believe that the research conducted complies to technical and ethical standards (in fact, this is quite an ethically indifferent research). I also believe that the described research is repeatable.

Validity of the findings

The findings of the research are in general relevant and to my opinion also novel. Also the conclusions are well stated and data has been provided. I also do believe that the approach that I would summarize as “getting the angle and then finetuning the base” may bring good results.

However, I do have several comments regarding the experiments and results:
1) The text contains quite a discussion regarding the achievable precision of measurement; however, I strongly believe that the achievable precision depends heavily (besides the optical and mechanical features of the stup) on the stereo base, and on the measured distance. Therefore, I believe that the paper should address this “parametrically” with taking into account these features. This is addressed in the study the paper refers to ([11]) but “this“ paper should be self-contained.
2) In Table 5 (and other measurements) it is clear that a “huge” difference exists between the “theoretical” expectation and real measurements. While this is explained in the paper “somehow”, to my opinion this rather indicates that the theoretical model does not work too well. Perhaps it would be worthwhile to include features of the setup in the theoretical model to make it more realistic.
3) I believe that as the “step” in the stereo base, which is 0,001m in this case, is too large as this, given the measured distance e.g. 800m, means step in the distance approximately of 800/0.25*0.001 = 3.2m is I understand it correctly, which then may complicate the measurement process. I believe that smaller step may be better. Would e.g. some subpixel identification of the position of the objects/markers in the images work better in the measurements?

Additional comments

I believe that it would benefit the paper to compare the experimental results to some other measurement methods, in other words, to the state of the art. This may bring the answer to the question whether the proposed method is better than state of the art and whether it brings some scientifically feasible knowledge. This, I believe, must be include in the paper.

Cite this review as

Reviewer 3 ·

Basic reporting

The manuscript is generally written in clear English.

The introduction provides adequate context, explaining the motivation to develop a more precise image-based distance measurement system for engineering applications. However, the literature review, while comprehensive, could benefit from a more critical analysis to highlight what specific technical gaps in prior work are being addressed here, beyond general accuracy concerns.

Figures are appropriate and clearly labeled. Raw data have been summarized in tables, and measurement comparisons are provided.

The structure of the paper follows expected norms: abstract, introduction, methods, results, discussion, and conclusion are all properly presented.

For completeness on references, the authors should consider RealSense stereo depth cameras, e.g., Leonid Keselman, John Iselin Woodfill, Anders Grunnet-Jepsen, Achintya Bhowmik; Proceedings of the IEEE Conference on Computer Vision and Pattern Recognition (CVPR) Workshops, 2017, pp. 1-10 (https://openaccess.thecvf.com/content_cvpr_2017_workshops/w15/papers/Keselman_Intel_RealSense_Stereoscopic_CVPR_2017_paper.pdf)

Experimental design

The study addresses an important problem: developing an image-based distance measurement device with engineering-grade accuracy. The research question is clearly defined and meaningful.

The experimental design is solid. The authors compare their developed image meter against a laser meter and a total station (used as ground truth). Measurements were performed on nineteen fixed targets under outdoor conditions, which is suitable for validating the approach.

The measurement procedures (angular adjustment, linear movement, pixel matching) are described in sufficient detail to allow replication. Tables systematically present measurements, error margins, and comparisons between theoretical and experimental results.

There is a genuine attempt to address mechanical inaccuracies (e.g., movement precision, electromechanical system errors) and to explain how these impact measurements.

However, there is no mention of environmental conditions (lighting, weather) during measurements, which could affect image processing and introduce additional variability.

The error sources are acknowledged but not quantified separately (e.g., mechanical vs. software errors). Future studies might benefit from more controlled isolation of these factors.

Validity of the findings

The findings seem valid overall. The developed prototype achieved an average error of 1.24% across all 19 measurements, with better performance at shorter distances.

The comparison with theoretical predictions is useful and shows that while theoretical error margins were lower, the practical system still performs commendably.

The study correctly acknowledges that mechanical inaccuracies limit the current performance.

The conclusions are properly linked to the experimental results and stay within the scope of what the data support. The limitations are candidly discussed, notably the amateur-grade mechanical system, and future improvements are suggested.

Below are some points for strengthening the validity:

Statistical analysis is minimal. Including standard deviations, confidence intervals, or hypothesis testing could strengthen the findings.

While the average error is reported, the distribution of errors (e.g., error variance with distance) could be better visualized or analyzed.

The fact that 11 out of 19 measurements outperformed the laser meter is a good point but could use more emphasis (e.g., statistically significant improvement?).

Additional comments

Strengths:
Novel mechanical design combining angular and linear camera movements.
Strong engineering motivation and practical relevance.
Clear and transparent reporting of results.
Acknowledgment of limitations and practical challenges.

Areas for Improvement:
More detailed discussion on environmental factors influencing performance.
Expand statistical analysis (error variability, confidence intervals).
Better isolation or characterization of mechanical versus software contributions to error.
Potential future work could explore calibration methods to reduce hardware-induced errors.

Cite this review as

---

## Round 0.2 · accepted · Accept

Many thanks to the authors for their efforts to improve the work. This version addressed the reviewers' concerns successfully. It can be accepted now. Congratulations!

·

Basic reporting

In the revised version of the article, the overall quality has improved. A prototype of a measurement device that is independent of pixel errors was developed, demonstrating that the method for precise distance determination using image processing is both effective and reliable.

Experimental design

n the experimental study, distances were measured using a total station and treated as the reference values. The research question is clearly defined and meaningful, and the study features an original design that effectively integrates mechanical control with vision-based processing.

Validity of the findings

The results are presented with clear and transparent error metrics, and the method is evaluated over a broad range of distances, including long-range measurements.

Reviewer 3 ·

Basic reporting

The review comments were adequately addressed in the revised version.

Experimental design

The review comments were adequately addressed in the revised version.

Validity of the findings

The review comments were adequately addressed in the revised version.

Additional comments

The review comments were adequately addressed in the revised version.

Cite this review as